# Aquaphotomics—From Innovative Knowledge to Integrative Platform in Science and Technology

**DOI:** 10.3390/molecules24152742

**Published:** 2019-07-28

**Authors:** Jelena Muncan, Roumiana Tsenkova

**Affiliations:** 1Biomedical Engineering Department, Faculty of Mechanical Engineering, University of Belgrade, 11000 Belgrade, Serbia; 2Biomeasurement Technology Laboratory, Graduate School of Agricultural Science, Kobe University, Hyogo 657-8501, Japan

**Keywords:** aquaphotomics, water, light, near infrared spectroscopy, water-mirror approach, perturbation, biomeasurements, biodiagnosis, biomonitoring

## Abstract

Aquaphotomics is a young scientific discipline based on innovative knowledge of water molecular network, which as an intrinsic part of every aqueous system is being shaped by all of its components and the properties of the environment. With a high capacity for hydrogen bonding, water molecules are extremely sensitive to any changes the system undergoes. In highly aqueous systems—especially biological—water is the most abundant molecule. Minute changes in system elements or surroundings affect multitude of water molecules, causing rearrangements of water molecular network. Using light of various frequencies as a probe, the specifics of water structure can be extracted from the water spectrum, indirectly providing information about all the internal and external elements influencing the system. The water spectral pattern hence becomes an integrative descriptor of the system state. Aquaphotomics and the new knowledge of water originated from the field of near infrared spectroscopy. This technique resulted in significant findings about water structure-function relationships in various systems contributing to a better understanding of basic life phenomena. From this foundation, aquaphotomics started integration with other disciplines into systematized science from which a variety of applications ensued. This review will present the basics of this emerging science and its technological potential.

## 1. Introduction to Aquaphotomics

Aquaphotomics is a young scientific discipline introduced by Professor Dr Roumiana Tsenkova at Kobe University in Japan in 2005 [1,2,3,4,5]. The establishment of a new science came in response to the recognized need in the current state of art for a common platform that can provide integration of knowledge about the water structure and functionality coming from various disciplines and most spectroscopy fields. 

Water is the simplest compound and is made of two most common reactive elements. It covers more than 70% of the Earth’s surface, comprises almost 2/3 of human body and is the most abundant molecule of all living cells. From nano to micro, meso, and up to the level of galaxies—water is everywhere. Wherever it is found, there are many phenomena involving it for which the mainstream science still does not have an explanation for. In everyday lives water is the first association to the word “liquid”, and yet liquid water is such an atypical liquid—with behaviors so different from other liquids—that its properties are called “anomalies”. This behavior stems from the capacity of water molecules for hydrogen bonding; if it did not exist, water would be a rather uninteresting material and our world would most likely look profoundly different. The hydrogen bonds connect the water molecules into a dynamic network; a water molecular network, or in other words—into a very complex water molecular system. 

The past two decades have seen much progress in water science. Due to the significant role it plays in biological systems, water has received considerable attention. Many interesting phenomena where water is a key player stimulated research across disciplines, revealing the significance of water structure and consequently its functionality in properties of materials or processes such as wettability [6], biocompatibility [7,8,9], cell communication and carcinogenesis [10], DNA structure [11], molecular recognition and communication [12], protein stability [10,13], membrane stability and survival in desiccated state [14], mechanical properties such as kernel hardness [15] or mechanical behaviors such as curling of the plant stem [16]—to list a few. Spectroscopy methods such as X-ray, infrared spectroscopy (IR), THz spectroscopy, near infrared (NIR) spectroscopy and others, using light as a probe, proved to be especially valuable tools for water studies and have contributed immensely to elucidation of various aspects of water systems. In general, water-light interaction over the entire electromagnetic spectrum, significantly contributed to a better understanding of water molecular systems [5]. 

Water molecules absorb radiation over the entire range of the electromagnetic spectrum (Figure 1). In contrast to mid- and far-infrared, where water strongly absorbs, allowing analysis of only very thin samples, in the NIR part of the spectrum, water absorption is much weaker, therefore offering the possibility of analyzing thicker samples and objects rapidly, in a completely non-destructive and non-invasive manner, and with none or little sample preparation. Using light of the NIR range, it is very easy to acquire spectral data of various aqueous and biological systems in real time without disruption of their state and dynamics. Near infrared spectroscopy thus offers a unique window of opportunities to observe the water molecular network as a scaffold—a matrix of every system of which it is an intrinsic part of—in relation to all other contributing elements and factors shaping the system structure, state and resulting dynamics—without any disruptions.

Aquaphotomics as a science was laid on a foundation provided by near infrared spectroscopy [5]. The breakthrough knowledge regarding the importance of water stemmed from the observation that NIR spectral data for milk of healthy and dairy animals with mammary gland inflammation (mastitis) differed at water absorbing bands (1440 and 1912 nm) [18,19,20,21]. The presence of disease in an organism influenced many biomolecules (fat, lactose, proteins, etc.); these changes were subtle and sometimes not even visible in the spectra at absorbance bands related to those compounds. However, all these components exerted an influence on the water structure, and this cumulative effect was observable and measurable at multiple water absorbance bands corresponding to different water molecular species. In other words, the water molecular network changed when the composition of the aqueous system was altered, and this was reflected in water spectral pattern. 

This innovative knowledge changed the approach in spectral analysis and paved way for the development of aquaphotomics. Changes in the water spectrum accurately and sensitively reflect the changes of water molecular species, hydrogen bonding and charges of the solvated and solvent molecules. In liquid water, each water molecule forms bonds with neighboring molecules, and can also establish dipoles and induce dipole interactions with other molecules, which gives the water molecular systems a heterogeneous character responsive to physical and environmental conditions [10]. Specific water species such as free water molecules, dimers, solvation shells and others contribute to the water spectrum in a very distinctive manner [5]. The water on a molecular level behaves as a collective mirror—its spectrum depicts changes as a response to all internal and environmental perturbations [5,22,23]. Rich experience acquired during many years resulted in a big database of spectra acquired under various perturbations, which revealed information regarding the water molecular system dynamics and the functionality of water in bio-aqueous systems [5], supporting the recognition of water as an active molecule and a central player in living processes [10,24]. 

The aim in establishing aquaphotomics as a science on its own, came in response to the recognized need in the current state of art in “omics” disciplines. Despite huge contributions of genomics, proteomics, metabolomics, transcriptomics and etc. to the comprehensive understanding of the principles underlying basic living functions, they all have an approach focused on single molecules and involve extraction procedures with the sample disruption. Biological systems can be studied using a non-destructive and integrative approach based on aquaphotomics, i.e., the interaction between water and biomolecules in which spectroscopic techniques combined with multivariate analysis represent a powerful tool.

Therefore, aquaphotomics aims at integrating and systematizing the knowledge about water-light interaction into a complementary, novel “omics” discipline whose objective is the large-scale, comprehensive study of water, its structure and related functionality. The first step towards this goal is identification of all the absorbance bands corresponding to specific water species. In this way, by knowing what each frequency means in the terms of the water structure, the absorbance bands become like “letters” that could be used to describe the features of aqueous systems. Relating the observed spectral patterns (combinations of water absorbance bands and the intensities of absorbance at these bands) with the observed characteristics or behaviors of aqueous systems, will clarify the functionality of certain water species and allow for future descriptions of the system states and dynamics solely in terms of the water structure. For various systems under various perturbations, aquaphotomics aims to build an aquaphotome, a comprehensive database of water bands and spectral patterns which describe the system and can therefore be used for future evaluations.

Since its establishment in 2005, aquaphotomics has showed steady progress (Figure 2). From only eight articles published in the first year after it was first introduced, the influence of the general idea and change over the years in the approach of how water is seen and treated in spectroscopy field can be seen. If the current trend continues, the estimate is that in the year 2025, 500 research articles per year can be expected. Through fundamental research, aquaphotomics provided novel insights and a better understanding of the basic phenomena and the role of water. It stimulated research and development of novel signal processing and chemometrics methods for data analysis, and provided a novel common measurement platform for a variety of applications, which led to the development of novel sensing devices and instruments based on water-light interaction. 

In the following sections, the key ideas of aquaphotomics, which lead paradigm shift of water seen from passive to active component of bio-aqueous systems and how this affected the basis of novel measurement platform—will be explained. Together with the brief illustrations of major contributions to science so far and through the extensive but not exhaustive list of applications, an overview of the huge technological potential of aquaphotomics will be presented. 

## 2. Water Spectrum as a Source of Information

### 2.1. Water as a Sensor and an Amplifier: The Water-mirror Approach

The fundamental idea of aquaphotomics is that water works as a sensor. This principle is in aquaphotomics—usually expressed in the terms of water being a “collective mirror” (the water-mirror approach) [5,22,25,26]. Every aqueous system is a dynamic arrangement of a water molecular network, hydrogen-bonded between themselves and/or other constituents and influenced by perturbations. As a consequence of the strong potential of water molecules for hydrogen bonding, water changes its absorbance pattern every time it adapts to a physical or chemical change in the system itself or its environment. The spectral pattern extracted through the interaction of light and water hence can be used as an integrative marker or descriptor of the state of aqueous system. 

The aquaphotomics approach is complementary to the conventional spectroscopy approaches. In most of the NIR-IR spectroscopy studies, the water absorption bands are considered to be masking the real information. For example, in order to measure proteins or sugars, the samples are usually dried in order to remove water and better observe the absorbance bands related to the structure of these biomolecules. In contrast, in aquaphotomics, the changes of the water spectral pattern are used as a source of information. The change in the concentration of a particular analyte is reflected in the changes of absorbance at several water absorbance bands, which are then used to build the prediction model. 

The water-mirror quality of water on a molecular level indirectly permits measurements of small quantities or structural changes of other molecules present in the aqueous system [27]. By tracking the changes in values of absorbance at water absorbance bands in the spectra of aqueous or biological systems, the information is extracted not only regarding the water structure but also of other components present in water, or the state of the system as a whole [4,5,27]. It should be mentioned that this property of water was recognized and utilized very early on in the field of NIR spectroscopy [28,29,30,31], but it was only with the development of aquaphotomics that the properties of water as a “collective mirror” were truly explored and the huge potential of it for understanding new phenomena and applications in aqueous systems, in biomeasurements, biodiagnostics and biomonitoring has been truly understood [5]. 

The fact that changes in concentrations of particular analytes affect many water molecules, and consequently affect many water absorbance bands, has a significant advantage. Traditionally, the quantification limit for NIRS is regarded to be the concentration of 5000 ppm (mgL^−1^) or 0.5% (*w*/*v*) [32]. This established limit for the traditional approach to NIRS analysis is based on the utilization of the absorbance bands of respective analytes directly. However, in aquaphotomics, water absorbance bands are used for indirect quantification. The comparison of different approaches— traditional and aquaphotomics and the resulting accuracy of quantification—was performed in one of the proofs of the concept works concerned with the measurement of concentration of polystyrene particles in water [33]. When the first overtone of water (i.e., the aquaphotomics approach) was used to develop a quantification model for polystyrene particles in aqueous suspension (1–0.0001%), the measurements achieved high accuracy—even in the case of very low concentrations. However, when the traditional approach was applied and measurements were based on the polystyrene band near 1680 nm (C-H stretching from aromatic C-H (2ν) [34]) a decrease in the concentration of particles led to a substantial decrease in prediction accuracy. Therefore, the two approaches are not equivalent. The possibility of detecting and measuring even low concentrations of analytes—lower than traditionally accepted limit for NIRS—is a result of the different principle of measurement. In all aqueous systems, every molecule of analyte is hydrated with an abundance of water molecules, which adapt to its structure and rearrange, creating a variety of different water molecular species that can be observed based on their respective absorbance bands in the NIR region. Since many water molecules are involved in the hydration of just one molecule of analyte, the water not only acts as a sensor, but also as an amplifier. This means that in aquaphotomics, instead of measuring analytes directly, the information about their concentration is obtained indirectly by measuring changes in always abundant solvent molecules which provides better detection and quantification than it was traditionally assumed the capability of NIRS technique.

Aquaphotomics can thus provide detection and quantification of analytes—even when they are not absorbing near infrared light—and when they are present in low concentrations [5,26,35]. In addition, this approach offers the possibility of using the same water spectra as a source of information about multiple analytes, thus enabling simultaneous measurements of many analytes [5,25,36]. 

### 2.2. Water Matrix Coordinates (WAMACS) and Water Spectral Pattern (WASP) 

In the NIR region, the water spectrum shows four main bands located approximately around 970, 1190, 1450 and 1940 nm, which are attributed to the second overtone of the OH stretching band (3ν_1,3_), a combination of the first overtone of the OH stretching and OH bending band (2ν_1,3_ + ν_2_), the first overtone of the OH stretching band (2ν_1,3_) and a combination of the OH stretching and OH bending band (2ν_1,3_ + ν_2_), respectively [37] (Figure 3). All these main bands are a rich source of information regarding the water structure. However, despite having lower absorbances or being overlapped with absorbance bands of other molecules, it should be noted that from 400 to 2500 nm more than 500 water absorbance bands have been identified [38]. 

Traditionally, water bands in the NIR region around 1450 and 1940 nm have been used for the determination of water content, hydration state [39] and, in particular, the moisture content [40] in various fields (agriculture, food industry, medical and pharmaceutical science). 

Conventionally, only few symmetric and asymmetric stretching vibration assignments of water molecules are known in the first overtone of water OH stretching vibrations (Figure 4). This region with its broad band might look completely uninteresting to a classical spectroscopist, and it is often overlooked as informationally poor. In fact, for years, water has been described as the ‘greatest enemy’ of infrared (IR) and NIR spectroscopy on account of its dominant absorption. 

The changes in water spectra in response to any change of water molecular network are very subtle, and require a data mining approach. The recognition of a high information potential of water spectra stimulated the development of novel analytical methods and even new computing tools, in order to meet the needs of the aquaphotomics data analysis [26,41,42,43]. A step-by-step explanation of aquaphotomics analysis supplemented by analytical tools currently at disposal is provided in Tsenkova et al. (2018) [26]. With so many tools at disposal, the utilization of the richness of NIR water spectra extended its applications far beyond moisture determination. 

Through extensive experimental research and application of multivariate analysis, the abundance of water absorbance bands in near infrared region was discovered [38]. In the area of the first overtone of water, 12 water absorbance bands corresponding to specific water molecular species were uncovered [5] (Table 1). These 12 absorbance bands, named water matrix coordinates (WAMACS), were found to be consistently important in spectral analysis of different aqueous and biological systems, and under different perturbations. Table 1 provides assignments for the WAMACS of the first overtone of OH stretching vibrations, based on the original publication as a source [5]. This list is only a small part of the full-scale list of water absorbance bands, which is far from being completed, and is continuously expanding as the science progresses. For example, in the region of C5 water matrix coordinate, recent work discovered a subpopulation of quazi-free water molecules—water molecules confined in the local field of ions (1396 to 1403 nm) [44]. 

Aquaphotomics starts to build up a “water vocabulary” where the “letters” are the water vibrational frequencies bands (WAMACS) and the water spectral patterns (WASP) are the “words” identifying different water spectral patterns and their relation to functions and phenomena in order to translate findings of water between different disciplines. More information about aquaphotomics terminology are reported in [26]).

The water absorbance spectral pattern WASP is usually presented by aquagrams [46]. There are different types of aquagrams [26]. The simplest form is a classical aquagram—a radar chart that displays normalized absorbance at selected water absorbance bands. For the first overtone of water, the axes of the aquagram are usually based on previously discovered 12 WAMACS. The normalized absorbance is calculated as follows:
$$A_{\lambda }^{\prime }=\frac{A_{\lambda }-\mu_{\lambda }}{\sigma_{\lambda }},$$
where $A_{\lambda }^{\prime }$ is normalized absorbance value displayed on radar axis; $A_{\lambda }$ is absorbance after scatter correction (multiplicative scatter correction using the mean of the dataset as a reference spectrum or standard normal variate transformation); $\mu_{\lambda }$ is the mean of all spectra; $\sigma_{\lambda }$ is the standard deviation of all spectra; and λ are the selected wavelengths from WAMACS regions corresponding to the activated water absorbance bands. Water absorbance bands are considered “activated” if they are consistently found to be among highly influential variables in the outputs of aquaphotomics analysis. 

The aquagrams are visually very convenient tools that enable quick and comprehensive comparison of different systems or conditions of the same system by comparison of their WASPs.

### 2.3. Using Perturbation to Elicit Information 

One of the most spectacular discoveries from the early years of aquaphotomics development was the observation that the absorbance spectrum of water changed with consecutive measurements [1] (in aquaphotomics, this is called illumination perturbations) (Figure 5). From the example presented in Figure 5, it is evident that every subsequent spectrum after exposure to near infrared light is different. The perturbation in the form of absorbed photons of radiation over sequential illuminations adds the energy to the system and changes the water molecular network. 

Apart from the most affected bands being located around 1410 and 1488 nm, the subtle troughs and shoulders can be observed throughout the spectra. The bands found to be affected by illumination are located at 1344, 1360, 1376, 1382, 1410, 1418, 1472 and 1482 nm—all within the ranges of 12 WAMACS and corresponding to different water species. These changes in water spectra in response to perturbation due to light serve as a source of additional information. For example, the first publications that presented the influence of light on water spectra, utilized the illumination as perturbation of prion protein solutions in order to discover differences in the functionality of different protein isoforms [1,47]. As the solution evolved with time, the frequencies of the various intramolecular vibration modes fluctuated due to a changing interaction between molecules. Out of three isoforms, only the solution of protein with bound copper ions consistently showed less bulk water despite the light perturbation suggesting it was the most stable form—a finding consistent with the published data.

Similarly to near infrared light, another aquaphotomics study that explored DNA mutation products, showed that exposure to ultraviolet (UV) radiation leads to changes in water spectral pattern of DNA solutions [48]. In addition, this study also showed that it was possible to measure the dose of exposure to irradiation with high accuracy (Figure 6). The regression vector of the developed model for prediction of the irradiation dose (Figure 6B) shows that UV light causes changes of absorbance at C5, C7, C8, C9, C10 and C11 water matrix coordinate. Comparing Figure 5 and Figure 6, the similarities between the influence of NIR and UV light on water spectral pattern can be observed. 

The illumination affects the water spectra similar to the temperature; it creates more free water molecules, which are then available to “scan” the rest of the water system and interact with its components. From this interaction, new information can be gained, as explained in the example above. 

Similarly to perturbation by light, intentional perturbation by temperature is often used in aquaphotomics as it is possible to use temperature dependent NIR spectra to obtain structural and quantitative information of the aqueous systems [42,43,49,50]. For instance, temperature perturbation was employed to study structural changes of ovalbumin as a model protein in aqueous solutions [51]. Two-dimensional correlation NIR spectroscopy and Gaussian fitting were adopted to investigate the variation of different water species and the sequences of the changes in the structure of protein during gelation. The results showed that in the gelation of protein, the change of S_2_ water species (water species with two hydrogen bonds) follows the same phases as the protein; it maintains the stability of the protein in native and molten globule states, while weakening of the hydrogen bond in S_2_ caused by high temperature resulted in the destruction of the hydration shell and led to ovalbumin clusters to form a gel structure.

In another work, water was used as a probe to quantify glucose in aqueous glucose solutions and human serum samples [52]. Spectral changes of water were captured from the temperature dependent NIR spectra using multilevel simultaneous component analysis (MSCA). The correlation coefficient for the temperature model was higher than 0.99, and that of the concentration of glucose were 0.99 and 0.84 for aqueous solutions and serum samples, respectively. Even if the changes in the spectra of water caused by temperature or concentration are very subtle, chemometrics provided techniques for the solution of this problem [50].

### 2.4. Water as a Biomolecule and Water Spectral Pattern as a Collective Biomarker

There are a hundred times as many water molecules in our bodies than the sum of all the other molecules put together. The most abundant molecule in the cell is water. Most biological processes involve water, and the interactions of biomolecules with water affect their structure, function and dynamics [10,24,53,54]. In the last decade, important advances have been made in our understanding of the factors that determine how biomolecules and their aqueous environment influence each other. 

In the field of near infrared spectroscopy, however, water is still not considered a molecular network or a biologically relevant matrix, which originates from the general opinion still dominant in life sciences that water is an inert, passive medium. The state of art is so that living processes are described in terms of genes, DNA, proteins, metabolites or other single biomolecules acting as entities isolated from water [53] (Figure 7a). In nature, there are no isolated biomolecules and water is not only the native environment in which all biological processes occur, but also an integral part of all of biological processes [10,24,55,56]. 

In aquaphotomics, the water spectral pattern is considered as the main source of information. This offers two advantages when analysis of biological systems is performed. First, by focusing on water absorbance, simultaneous measurements of several analytes is possible, and second, which is far more important, the cumulative effect of different biomolecules on water matrix offers opportunity of using water spectral pattern as a novel biomarker. In most conventional spectroscopy studies, quantitative models are made for each separate component to be used to diagnose a system, where combining the models multiplies the errors—thereby producing inaccurate results. In aquaphotomics, despite possibility of measurements of the individual components, using the water spectral pattern as an integrative, global marker provides much more information about the studied systems, because it includes the effect of all components in the system—even the ones that, at the moment, current science does not identify as important and contributing to the system functionality (or disfunctionality, as is the case of diseases). 

In aquaphotomics, the ‘functionality’, the biological state, the biological reaction to a change (dynamics) of the bio aqueous system is the key (focus, objective), instead of the presence of individual molecules. Specific water molecular structures (presented as water spectral patterns) are related to the status, dynamics and ‘function’ of the bio-aqueous systems studied, thereby building an aquaphotome—a database of water spectral patterns correlating water molecular structures to specific ‘perturbations’ (disease state, contamination state, reaction to light, change in temperature, and so on). 

The simple shift in perspective of what water is in biological systems offers novel insights and explanations of certain biological processes or phenomena. For example, DNA damage was detected through changes in the water spectral pattern as nonirradiated and UVC-irradiated DNA solutions were successfully distinguished in the 1488–1543 nm range, corresponding to the first overtone of water [48].

## 3. Aquaphotomics—Innovative Knowledge Leads to Innovative Applications

Being rapid and non-destructive, NIR spectroscopy is a powerful technique whose horizons have been further expanded by aquaphotomics. Since its establishment, aquaphotomics has grown into a multidisciplinary scientific field, encompassing many research areas and providing a common measurement platform for many applications. Using near infrared spectroscopy in aquaphotomics—in comparison to using light of other frequencies—does offer significant advantage of non-destructive evaluation of aqueous systems, which is of special significance for not only exploration of biological systems, but offers immense potential for biodiagnosis and biomonitoring. This region is furthermore an excellent tool for water observation, which provides an enormous amount of information about water molecular structure [5,58]. Numerous NIR spectra can be obtained in various conditions and states of the systems (under different perturbations)—all in real time. 

The work in such a wide variety of applications, with different systems in different conditions led to two significant breakthroughs in aquaphotomics. The first breakthrough is that water spectral pattern can be used as a collective, integrative biomarker—a descriptor of a system’s state [4]. The second one is the discovery that the water spectral pattern is related directly to certain functionality of the system. While the first breakthrough is of major significance for applications and provides a novel measurement platform, the second one leads to innovative knowledge of many phenomena. The next sections will illustrate the significance of both. 

Contrary to the common understanding of overtone spectroscopy (100 to 1000 times lower absorbance than in the mid-IR range), it has been shown that even very small concentrations of the solutes could be measured with NIR spectroscopy if the aquaphotomics water-mirror approach is applied. Changes in the absorption spectrum of liquid water were used for quantification of the solutes present in water, even when the solutes did not absorb NIR light at all [30,35]. For instance, using very robust experimental design, Gowen et al. performed comprehensive aquaphotomics analysis of aqueous salts solutions (NaCl, KCl, MgCl_2_, AlCl_3_) with the aim of establishing limit of detection [35]. This research demonstrated that the best region for the prediction of salt concentration was the first overtone of water, attaining the prediction error of 500-800 ppm. Similar detection limit (1000 ppm) was reported in a research study that explored quantification of different metals (Cu(II), Mn(II), Zn (II) and Fe(III)) in aqueous HNO_3_ [59], while another work reported successful prediction of HIV virus concentrations in plasma with the standard error of 23 pg/ml (ppb level) [60]. The water-mirror, indirect approach enables measurements of concentrations previously thought impossible to be measured with NIR spectroscopy at ppm and even at ppb levels under certain experimental conditions [23,25,33,35,59,60,61,62]. However, if we look beyond the measurements of individual solutes, what these results illustrate is the sensitivity of water molecular network to the changes in its components. The successful applications list measurements of acidity, pH [63] and effects of mechanical filtration on pure water [64]. Introducing water spectral pattern as an integrative marker represents one step forward from the detection of individual contaminants in water quality monitoring [65] or measurements of single, individual biomarkers in disease diagnostics [4]. 

This concept is radically novel, because it shifts the perspective of the definition of water quality by a set of physico-chemical and microbiological parameters to the definition of water quality as a water spectrum within some defined spectral limits. The same is true for disease diagnostics, which for many of diseases, especially in the early stage of development, works with very low concentrations of biomarkers in body fluids or does not even have reliable biomarkers. The spectrum of aqueous system integrates the influence of all single markers into one integrative, holistic marker which is a result of cumulative effect of many components and can easily be monitored in real time. The applicability of the proposed concept was evaluated in water quality monitoring [65], food quality monitoring [66] and biodiagnostics [67,68]. Using water as a biomarker, the information on the health status of any organism can thus be acquired in real time and non-destructively, allowing the continuous in vivo monitoring of the same sample. 

In plant biology studies, aquaphotomics provided a methodology to follow the impact of a virus infection based on tracking changes in water absorbance spectral patterns of leaves in soybean plants during the progression of the disease [69,70]. Compared to currently used methods such as enzyme-linked immunosorbent assay (ELISA), polymerase chain reaction (PCR), and Western blotting, aquaphotomics was unsurpassable in terms of cost-effectiveness, speed, and accuracy of detection of a viral infection. The diagnosis of soybean plants infected with soybean mosaic virus was done at the latent, symptomless stage of the disease based on the discovery of changes in the water solvation shell and weakly hydrogen-bonded water which resulted from a cumulative effect of virus-induced changes in leaf tissues. A similar study reported the detection of begomovirus in papaya leaves with an aquaphotomics approach [71]. Tracking the cumulative effect of various, most likely unknown, biomarkers of viral infection in leaves provided grounds for successful, early diagnosis based on aquaphotomics principles. 

Similarly, different water spectral patterns were found in leaves of genetically modified soybean with different cold stress abilities [69]. This research on the discrimination of soybean cultivars with different cold resistance abilities has proven that resistance to cold stress can be characterized by different water absorbance patterns of the leaves of genetically modified soybean. Different genetic modifications resulted in a multitude of bio-molecular events in response to cold stress, whose cumulative effect was detected as a specific water spectral pattern of leaves; i.e. the higher the cold resistance, the higher was the ability of the cultivar to keep the water structure in less-hydrogen bonded state, providing a supply of “working water” in the conditions of decreased temperature. 

In another study, aquaphotomics was applied for exploration of the extreme desiccation tolerance i.e. the ability of some plants—called resurrection plants—to survive extremely long periods in the absence of water and then to quickly and fully recover upon rewatering [72]. Application of aquaphotomics to study one such plant—*Haberlea rhodopensis*—during dehydration and rehydration processes, revealed that in comparison to its biological relative—a non-resurrection plant species, *Deinostigma eberhardtii*—*H. rhodopensis* performs fine restructuring of water in its leaves, preparing itself for the dry period. In the dry state, this plant drastically diminished free water, and accumulated water molecular dimers and water molecules with four bonds (Figure 8). The decrease of free water and increase of bonded water, together with preservation of constant ratios of water species during rapid loss of water, was found to be the underlying mechanism that allows for the preservation of tissues against the dehydration-induced damages and ultimately the survival in the dry state.

In the medical field, aquaphotomics was proposed for in vivo therapy monitoring of topical cream effects [73,74], for monitoring of dialysis efficacy [67] and diagnosis of several diseases: cancer [67], diabetes and coronary heart disease [75]. These applications utilize the concept of a water spectral pattern as an integrative biomarker that offers significant advantage compared to traditional ways of therapy monitoring or diagnostic practices in medicine. For example, monitoring dialysis efficacy is a particularly challenging task that relies on discrete sampling and measurements of only several uremic toxins out of more than 80 currently recognized that contribute to the uremic syndrome (Figure 9). The NIRS method has already been proposed to measure urea in spent dialysate [76]. However, urea is only a single marker and its concentration decreases during dialysis, making the detection harder. By using aquaphotomics approach, individual component measurements were replaced by process monitoring [67]. Instead of measuring waste materials in spent dialysate, their cumulative effect on the water matrix was measured as water spectral pattern changes during the dialysis. In another words, individual component measurement was replaced by monitoring of the process. The water spectral pattern of spent dialysate averaged for all patients after 5, 45, 90 and 135 min of treatment presented as the aquagram in Figure 9 showed, as the therapy progressed there was an increase of free water molecules (1398 and 1410 nm: C5 WAMACS) in the dialysate. In this way, the efficacy of dialysis can be assessed in a simplified way by tracking the changes of the respective dialysate water spectral pattern. The advantage of such an indirect approach of biomonitoring can also be extended to biodiagnostics as the water spectral pattern captures the information regarding all biomolecules that change with the disease—even the biomolecules current science is not aware of. 

The works on mastitis [21,77,78,79,80,81,82] showed that as the various milk components change during the different stages of infection, they influence the water matrix of milk differently. The water spectral patterns of blood, milk, and urine of mastitic cows, revealed that the same water absorbance bands are activated in different body fluids in response to the presence of disease [81]. Similarly, physiological changes such as ovulation can be detected in various body fluids using the same principles such as in the Giant panda [68,83], in the Bornean orangutan [84], in dairy cows [85] and in mares [86]. 

Aquaphotomics made a significant contribution to the field of microbiology and food engineering by not only providing a fast and nondestructive analysis, but by contributing to better understanding of the mechanism of action of some microorganisms [87,88,89]. For example, probiotic, non-probiotic and moderate bacteria strains produced a unique water spectral pattern, as shown in aquagrams reported in Figure 10. Probiotic bacteria strains were characterized by a higher number of small protonated water clusters, and free water molecules and water clusters with weak hydrogen bonds [89]. The discovery that strong probiotic bacteria produced more free water and less hydrogen-bonded water species, i.e. they break water structures in a way comparable to an increase in temperature, provides novel insight on their mode of action. Moreover, aquaphotomics was able to distinguish a subdivision into two species within one bacteria strain, where conventional PCR analysis was not enough sensitive [90]. 

The aim of studying water interactions on a molecular level was to obtain a better understanding of the relationship between the water structure and a phenomena on a macro scale. For example, one of the novel studies related the sensory texture of apples with particular spectral pattern of fruits: mealy apples had water predominantly in a weakly hydrogen bonded state, while the opposite was true for juicy, firm apples [66]. Another study related the dehydration band (1398 nm) with physical damage in mushrooms [45]. Similarly, wheat kernel hardness was related to specific water absorbance bands (1366 and 1436 nm) [15]. Usually, food texture is not considered a property that stems from certain water structure; however, the above-mentioned works revealed that water structure change with texture. Further studies are needed to better understand the relationships between the water spectral pattern and pectin metabolism in horticultural products.

Interesting findings were obtained in applications of aquaphotomics for basic studies of interaction of biomolecules and water. For example, although many spectroscopic studies have been conducted on glucose, few studies have been carried out on the anomers of glucose despite the fact that spectra—as well as chemical and enzymatic reactions—depend on the specific molecular structure. What aquaphotomics study of glucose isomerism [91] found is that the absorbance band at 1742 nm possess the potential to distinguish glucose anomers qualitatively and quantitatively. What is conventionally regarded as the first overtone of the C-H stretching mode, was confirmed to not be related to glucose—but to water [92,93]. Through work in the field of protein-water interactions, aquaphotomics provided insight into their dynamics and the significant role water plays in their functionality. In a study of prion protein isoforms [47], aquaphotomics analysis of Mn and Cu prion isoforms in water solutions revealed that while binding of copper results in increased protein stability in water, the binding of manganese resulted in less stability—which led to fibril formation, responsible of neurodegenerative disease. The fact that the entire process of protein structural changes in aqueous systems can be monitored indirectly through the water absorbance pattern of the protein solution, was demonstrated in a study of amyloid protein—another protein involved in pathogenesis of neurodegenerative diseases [71]—as well as ovalbumin [51]. 

Aquaphotomics studies on water-material interaction hold great promise in understanding some of the very complex properties that are of interest for many applications, such as wettability or biocompatibility. A study concerned with investigation of an excellent wettability of titanium dioxide reveled the importance of water species ratios [6]. More recent studies exploring the state of water in hydrogel materials of soft contact lenses [94,95,96] revealed that the water spectral pattern holds information even about the state of polymer network and protein deposits on the surfaces of worn contact lenses. Other aquaphotomics studies showed how nanomaterials shape the water matrix, as in the case of fullerene-based nanomaterials that act as water structuring elements when present in very low concentrations [73,74,97]. In nanotechnology and nanomedicine, aquaphotomics could lead to novel findings due to the fact that with decreasing size, the available active surface interacting with water playing a significant role increases. 

Since its establishment, aquaphotomics has grown into a large, multidisciplinary scientific field, encompassing many research areas and providing common measurement platform for many applications. Table 2 provides an idea of possible fields of applications of aquaphotomics coupled to NIR spectroscopy. These works illustrate the great versatility of this technique and can hopefully inspire novel research and application ideas. 

## 4. Future Perspectives

With the theoretical and technological advancements in spectroscopy and data analysis techniques, the development of aquaphotomics as a new science led to development of a steady new knowledge base about water-light interaction and provided a common measurement platform that employs novel measurement principles. 

The future of aquaphotomics will be towards building up the aquaphotome database of WAMACs and WASPs for extensive number of systems in our life. It will embrace the rest of the “–omics” data in a collective manner to be used as complementary tool for further understanding new phenomena in science and for the development of feedback systems where the WASP will be the diagnostic tool and the respective individual “–omics” data will provide the information for the regulator in a feedback system to not only monitor and diagnose, but control processes including biological ones.

The innovative knowledge of importance of water as a biologically significant molecule (a biomolecule in its own right [53]) and of water-light interaction as a way of extracting information led the paradigm shift that places different demands on the development of sensing technologies. The advantage of being non-destructive, rapid and capable of comprehensive biomonitoring and biodiagnosis, based on utilization of water spectral pattern as new, more accurate and collective biomarker, aquaphotomics provides great potential to complement conventional technologies used to perform single tasks, while in others it may even lead to the replacement of current ones. 

The aquaphotomics based applications vastly extended the possibilities of spectroscopy and especially of the near infrared spectroscopy, while the ever improving sensor technology offers great prospects for high accuracy, real-life applications, being more cost-effective at the same time [134]. Presented here, aquaphotomics works demonstrate outcomes that are presumably just a glimpse of a much larger application potential.

## Figures and Tables

**Figure 1 molecules-24-02742-f001:**
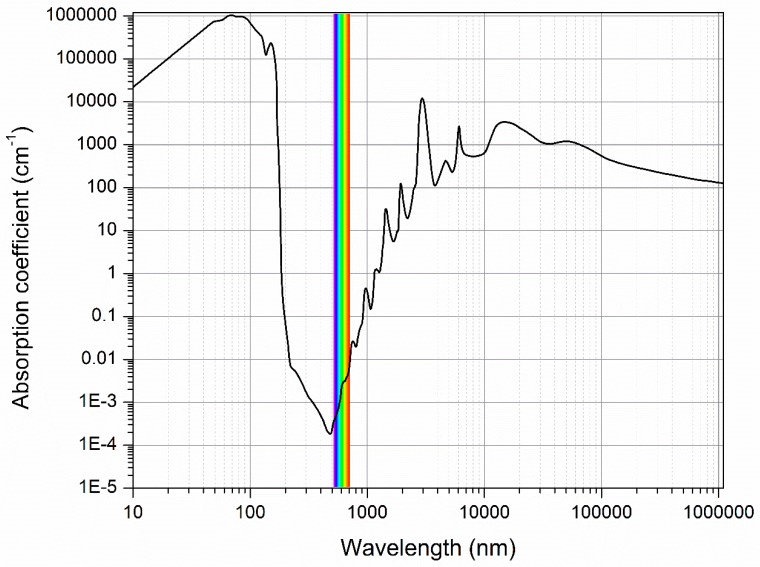
Water spectrum (double logarithmic plot), based on data from Segelstain [17].

**Figure 2 molecules-24-02742-f002:**
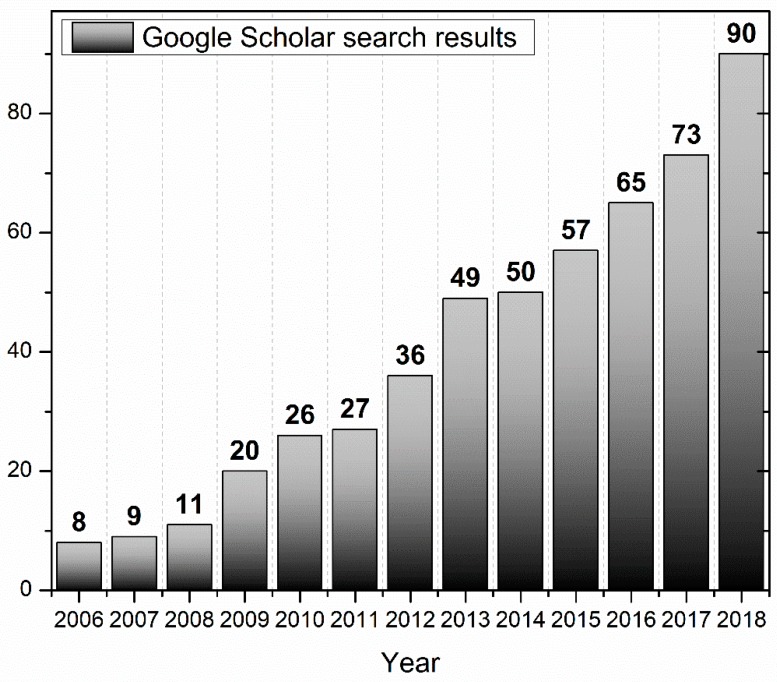
Number of articles published per year mentioning word “aquaphotomics” since 2006. The numbers are obtained using Google Scholar web search engine for articles and patents (excluding citations) containing word “aquaphotomics”.

**Figure 3 molecules-24-02742-f003:**
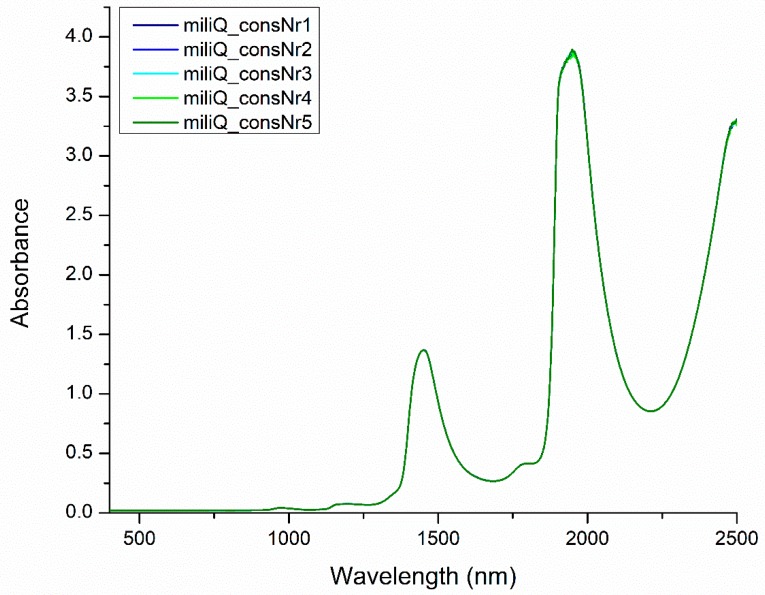
Spectra of pure water (produced by Milli-Q water purification system (Millipore, Molsheim, France) in the visible-near infrared region (400–2500 nm). Five spectra (miliQ_consNr1, miliQ_consNr2, miliQ_consNr5) presented in the figure were acquired by illuminating the same water sample five times consecutively.

**Figure 4 molecules-24-02742-f004:**
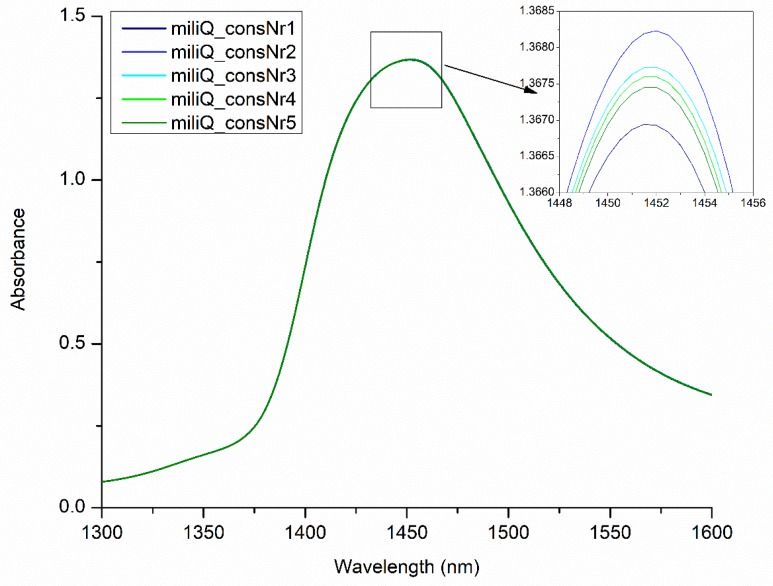
Spectra of pure water subjected to consecutive illuminations (the same spectra presented in Figure 3) in the area of the first overtone of water.

**Figure 5 molecules-24-02742-f005:**
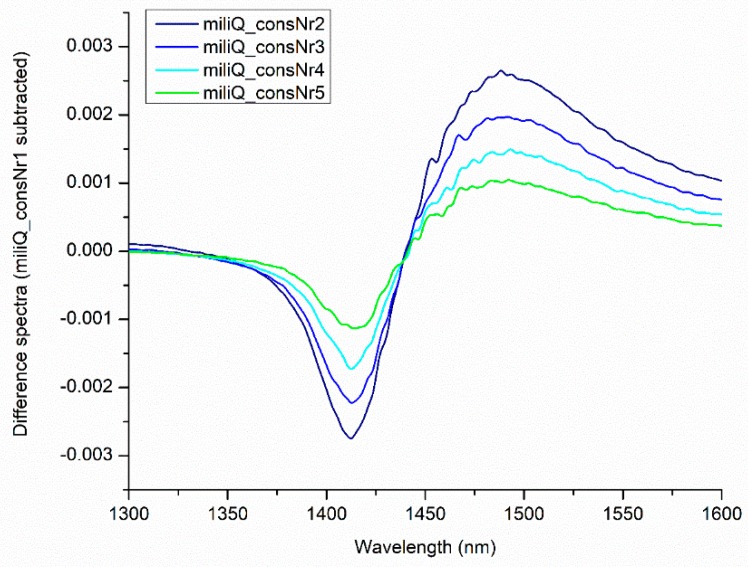
Consecutive illumination of water changes the near infrared spectra. Difference spectra calculated by subtracting the first consecutive spectrum from four subsequently acquired spectra under consecutive illuminations (the same spectra from Figure 3 and Figure 4), show that near infrared light changes the water spectral pattern.

**Figure 6 molecules-24-02742-f006:**
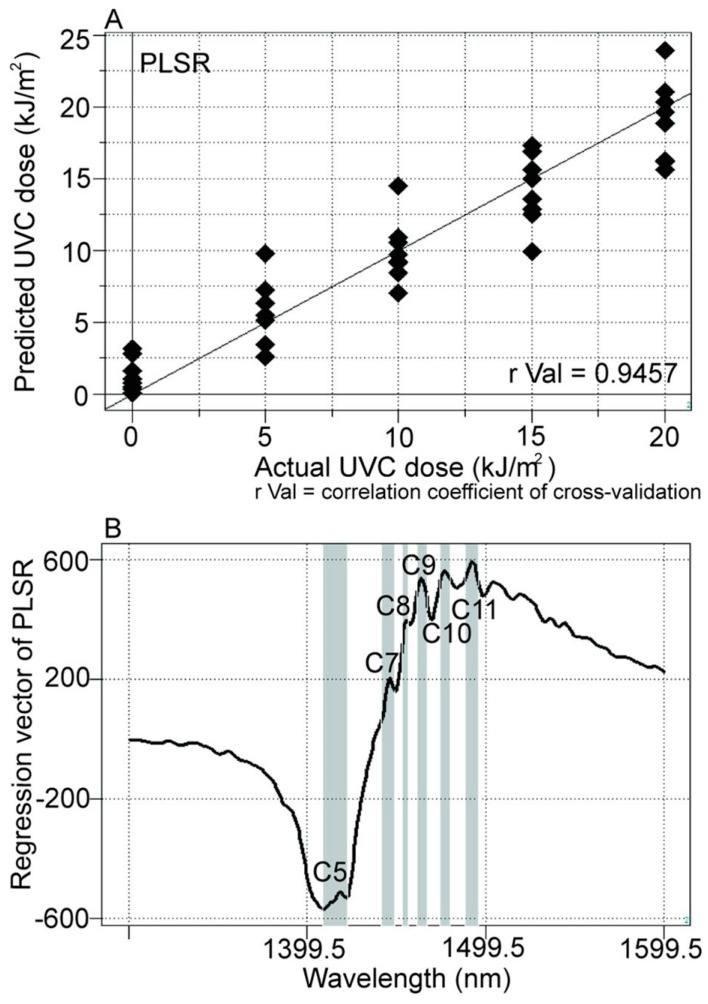
PLSR model for prediction of UV irradiation dose: (**A**) Y-fit curve showing relationship between actual and predicted values; (**B**) regression vector of PLSR model showing water absorbance bands affected by UV light perturbation [48].

**Figure 7 molecules-24-02742-f007:**
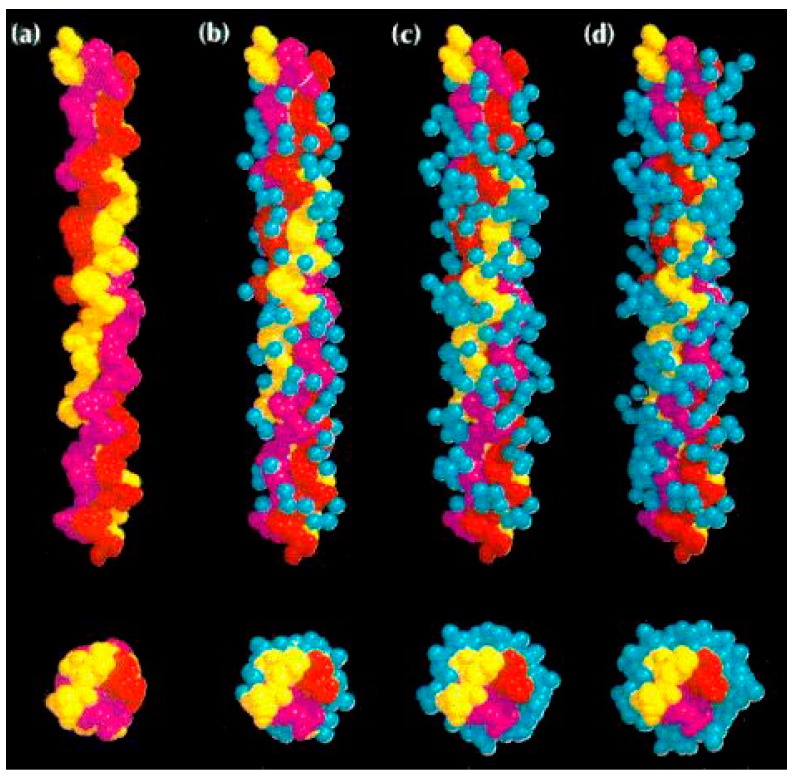
An example of a collagen peptide and its hydration shell: (**a**) in conventional science, biomolecules are usually represented only by this main chain on a black background, as if the biomolecular processes are happening in the vacuum; (**b**–**d**), a realistic picture, showing water hydration shells as an integral part [57] (Reprinted from Bella J, Brodsky B, Berman HM. Hydration structure of a collagen peptide. Structure 1995; 3:893–906, with permission from Elsevier).

**Figure 8 molecules-24-02742-f008:**
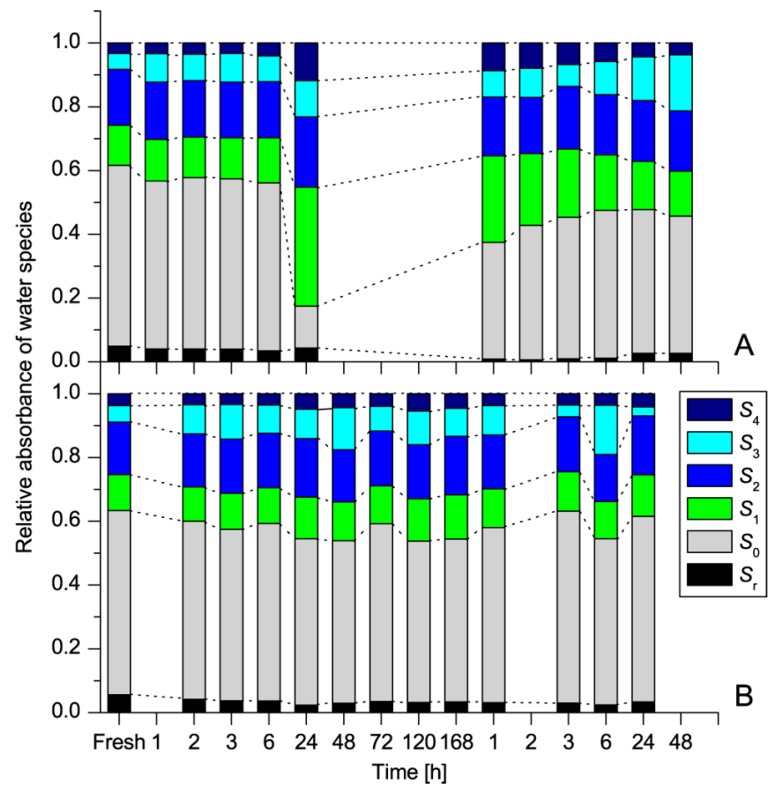
Dynamics of different water species (S*_i_* = water molecules with *i* hydrogen bonds, S_r_ = protonated water clusters) during dehydration and rehydration of *Haberlea rhodopensis* and *Deinostigma eberhardtii*. Relative absorbance of water species in *Haberlea rhodopensis* (A) and *Deinostigma eberhardtii* (B) during desiccation and subsequent rehydration [72].

**Figure 9 molecules-24-02742-f009:**
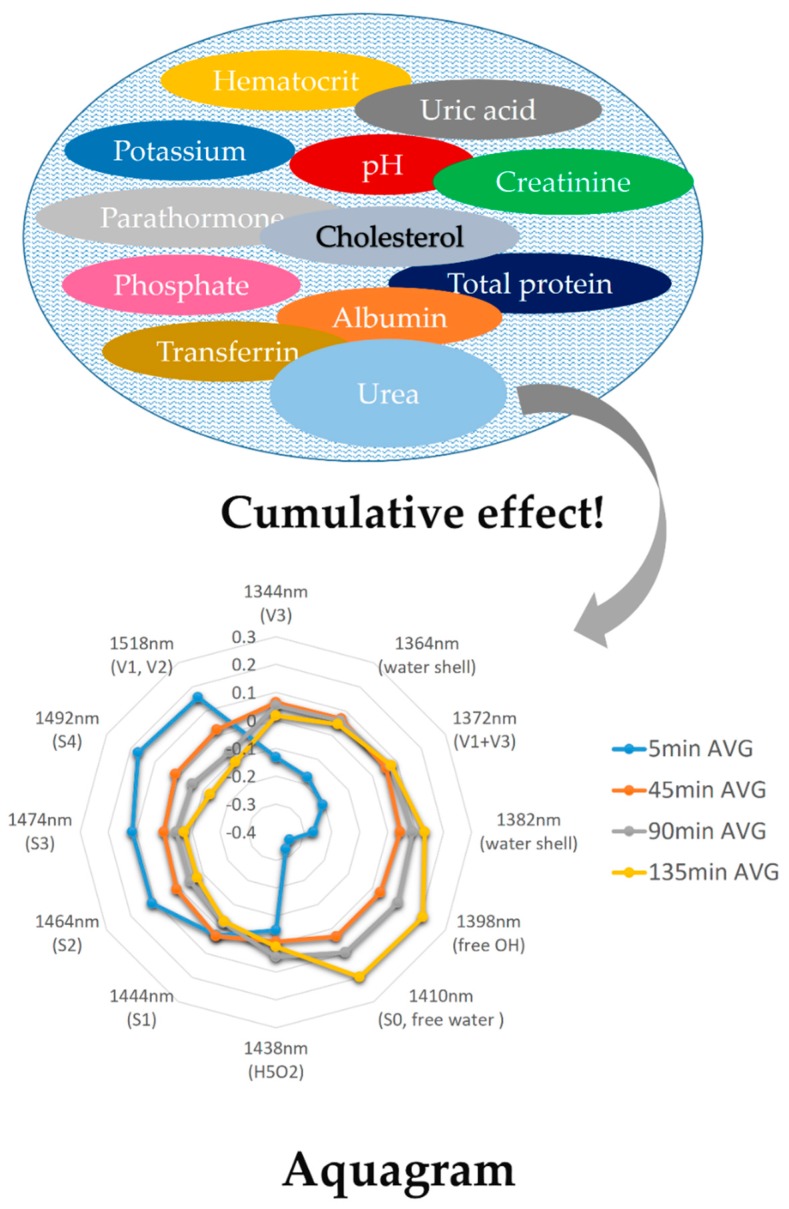
Water spectral pattern of spent dialysate presented on aquagram can be used as a marker of dialysis efficacy. Instead of measurements of different uremic toxins (of which there are more than 80), aquaphotomics provides measurement of their collective cumulative effect on water matrix of spent dialysate. [67].

**Figure 10 molecules-24-02742-f010:**
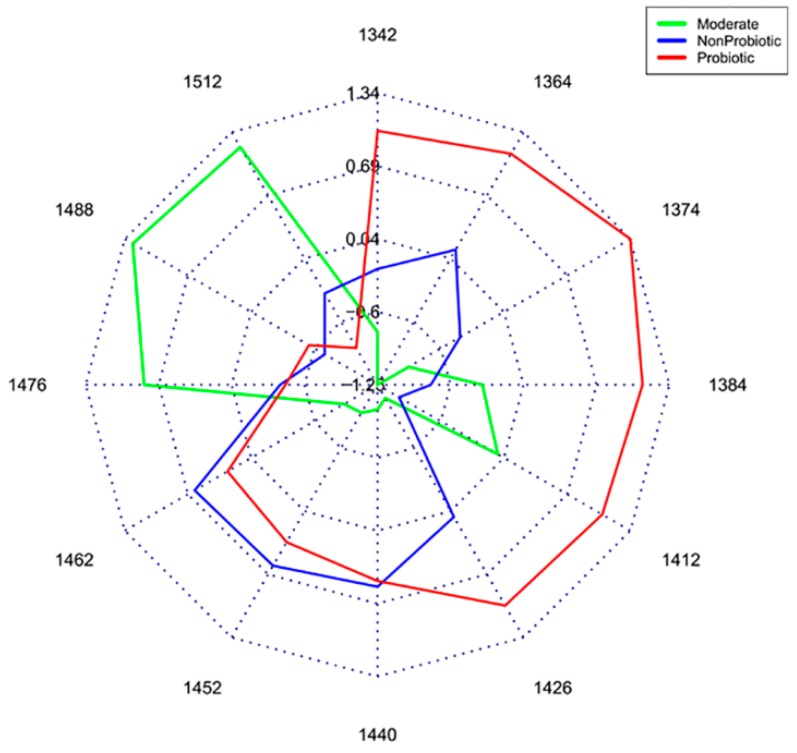
Aquagrams of culture media of groups of probiotic, moderate and non-probiotic strains. Average values of normalized absorbance values of the water matrix coordinates for each group are plotted on each wavelength axis [89].

**Table 1 molecules-24-02742-t001:** Water matrix coordinates in the area of the first overtone of water in the near infrared region (1300 to 1600 nm) (based on [5,44,45]).

WAMACS	Range (nm)	Assignment
**C1**	1336–1348	2ν_3_: H_2_O asymmetric stretching vibration
**C2**	1360–1366	OH-·(H_2_O)_1,2,4_: Water solvation shell
**C3**	1370–1376	ν_1_ + ν_3_: H_2_O symmetrical stretching vibration and H_2_O asymmetric stretching vibration
**C4**	1380–1388	OH-·(H_2_O)_1,4_: Water solvation shell O_2_-·(H_2_O)_4_: Hydrated superoxide clusters 2ν_1_: H_2_O symmetrical stretching vibration
**C5**	1398–1418	Water confined in a local field of ions (trapped water) S_0_: Free water Water with free OH-
**C6**	1421–1430	Water hydration band H-OH bend and O-H…O
**C7**	1432–1444	S_1_: Water molecules with 1 hydrogen bond
**C8**	1448–1454	OH-·(H_2_O)_4,5_: Water solvation shell
**C9**	1458–1468	S_2_: Water molecules with 2 hydrogen bonds 2ν_2_ + ν_3_: H_2_O bending and asymmetrical stretching vibration
**C10**	1472–1482	S_3_: Water molecules with 3 hydrogen bonds
**C11**	1482–1495	S_4_: Water molecules with 4 hydrogen bonds
**C12**	1506–1516	ν_1_: H_2_O symmetrical stretching vibration ν_2_: H_2_O bending vibration Strongly bound water
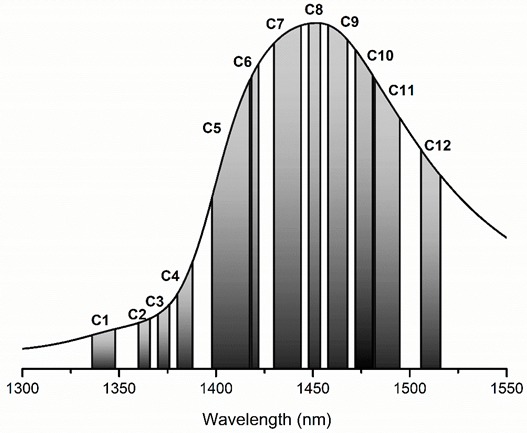		

**Table 2 molecules-24-02742-t002:** Aquaphotomics contribution: from fundamental research to various applications.

Application	Object of Study	Purpose	References
**Fundamental research**	Sugars	Quantification	[25,43,49,52]
	Glucose	Distinguishing anomers	[91]
	Salts	Quantification and influence on water spectra	[26,35,44,61,98]
	Acids	Quantification, accuracy of prediction depending on acidity	[99]
	Acids and pH	Quantification	[63]
	Ethanol	Quantification, structural analysis	[42,100,101,102,103]
	Methanol	Quantification	[98,104]
	Water-ethanol-isopropanol mixture	Quantitative analysis and the effect of temperature	[105]
	Water, methanol, ethanol and ethylenediamine mixture	Quantitative analysis and the effect of temperature	[106]
	Monoethylene-glycol	Quantification	[98]
	Metal ions	Quantification	[107,108,109,110]
	Near infrared light	Influence of consecutive irradiation	[1]
	UV light	Measurement of irradiation dose	[48]
	Temperature	Influence of temperature on water spectra	[42,43,111]
**Biomolecules**	Oligopeptides	Interaction with water – elucidating the structure, dynamics and function of proteins	[112]
	Prion proteins	Stability of protein structure as a function of metal binding	[47]
	Insulin	Fibrillation phases	[113]
	Albumin and γ-globulin	Quantification	[114]
	Albumin	Structural analysis and hydration properties	[115]
	Ovalbumin	Gelation of globular proteins	[51]
	DNA	Quantification and detection of mutation products	[48]
	Phospholipids	Structural analysis and effect on water	[111]
**Water**	Water contamination	Quantification of pesticides alachlor and atrazine	[62]
	Water contamination	Detection of contaminants based on salts as model systems	[35]
	Commercial mineral waters	Discrimination	[116]
	Ground water quality	Continuous monitoring based on water spectral pattern as a holistic/integrative marker	[65]
	Pure water	Influence of filtration process	[64]
**Food**	Honey	Adulteration	[117]
	Mushrooms	Detection of physical damage	[45,118]
	Milk	Components	[36]
	Wafer, coffee, soybean	Water activity and moisture content	[119,120]
	Perches (fish)	Discriminating between wild fishes and raised in the recirculation system	[121]
	Pork loin	Discrimination between fresh and spoiled meat	[121]
	Porcine muscles	Discrimination between fresh and thawed meat	[121]
	Cheese	Ripening process	[122]
	Cheese and winter melon	Influence of packaging material on ripening	[123]
	Salami	Influence of coating on ripening	[124]
	Packaging material	Influence of bioactive compound - propolis	[125]
	Apples	Sensory texture - specific mechanical and structural properties related to water spectral pattern	[66]
	Oilseed Rape	Stem rot detection	[126]
	Rice	Seed vitality	[127]
	Coffee	Roasting degree	[128]
	Wheat kernels	Hardness	[15]
**Materials**	Soft contact lenses: hydrogels	Discrimination of hydrogels with different water content	[95,96]
	Soft contact lenses: hydrogels	Discrimination of new and worn contact lenses	[94]
	Titanium dioxide	Wettability	[6]
**Environment**	Soil	Identification of soil type	[129]
	Water contamination	Monitoring	[65,130]
**Nanomaterials**	Fullerene based nanomaterials	Hydration properties	[73,74,97]
	Polystyrene	Quantification of particles in water solutions	[33]
**Microbiology**	Bacteria – metabolites	Contribution to NIR signal from cells and metabolites	[131]
	Bacteria - probiotic	Classification	[87,89]
	HIV virus	Detection and quantification	[60]
	Bacteria	Selection	[88]
**Cells and tissues**	Somatic cells in milk	Quantification	[21]
	Tissue (mice)	Native state of metals	[132]
	Tissue (mice)	*Ex vivo* discrimination	[133]
**Plant biology**	Soybean	Detection of mosaic virus infection	[70]
	Soybean	Ability to cope with cold stress in genetically modified cultivars; Detection of mosaic virus infection	[69]
	Resurrection plants	Peculiarities of water structure in leaves of anhydrobiotic organism	[72]
	Papaya leaves	*In vivo* detection of begomovirus infection	[71]
**Animal medicine**	Mastitis in dairy cows	Disease detection	[21,77,78,79,80,81,82]
	Estrus detection in urine of giant panda	Finding water spectral pattern as biomarker, quantification of hormone	[68,83]
	Estrus detection in milk of cows	Ovulation period detection and monitoring	[85]
	Estrus detection in urine of Bornean orangutan	Ovulation period detection and monitoring	[84]
	Estrus period detection using serum in mares	Detection of oestrus, metestrus, and diestrus in mares,	[86]
**Medicine**	DNA mutation products	Detection of DNA damage, quantification of damage products	[48]
	AIDS	HIV virus detection	[60]
	Serum	Serum based diagnosis (diabetes, coronary heart disease)	[75]
	Prion protein disease	Mechanism of disease	[47]
	Skin cream effects	Therapy monitoring	[73,74]
	Dialysis efficacy	Monitoring of spent dialysate	[67]
	Colorectal cancer	Diagnostics based on serum and urine	[67]

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
