# Peer review of "Aquaphotomics—From Innovative Knowledge to Integrative Platform in Science and Technology"

_molecules, 2019, doi:10.3390/molecules24152742_

Round 1

Reviewer 1 Report

The manuscript ID: molecules-551065 entitled “Aquaphotomics - From Innovative Knowledge to Integrative Platform in Science and Technology” deals with the topic of aquaphotomics in a complete and exhaustive way.

However, the current paper is a little bit redundant, some concepts are repeated too many times (water is an enemy for NIR traditional analysis, water is a molecular network…., water is a mirror of a biological system….).

In addition, the authors have used a too convoluted and, sometimes, pompous language. The sentences are too long, and the same concepts could be expressed in a simpler way.

It is suggested to quickly review, shorten and summarize the current manuscript, taking into account the suggestions reported in the attached file.

Reviewer 2 Report

Aquaphotomics is a nowadays established discipline, whose role in understanding simple and more complex phenomena mediated and "reflected" by water itself is continuously growing. In this context, the manuscript presented offers a clear overview of the genesis of the discipline and how it evolved into a mature field, together with a critical discussion of its main elements and some examples of application. 

For these reasons I do strongly recommend publication on the journal. Moreover, the manuscript is well written and each relevant issue is treated in depth, so prior to publication, I suggest the following revisions:

General: 

The manuscript needs a further reading to look for avoidable repetitions, misspells, typos and sentences that are sometimes hard to read.

Below a few of these corrections to be made are reported.

Lines 39-40: "still.......still". Try to avoid the repetition; also because the second "still" is superfluous. 

LIne 62: Change "infrared" to "mid and far infrared".

Lines 91-92: "specific....specific". Try to avoid the repetition.

Line 124: "aquaphotomics" instead of "the aquaphotomics".

Lines 167-168: should be "considered to be masking".

Line 177: "NIRS [33]". "NIRS" should be removed.

References: They should be reported in a consistent way (see, e.g., Ref.8 vs Ref.9)

Reviewer 3 Report

Dear Authors, I personally appreciated your comprehensive review on Aquaphotomics; the work is well presented and written with a particular aim of explaining this new discipline to interested researchers. 

I suggest the publication on Molecules after some minor comments:

Line 143: I suggest changing the title of section 2.1, to underline the fact that the water mirror approach will be presented. For example, the title can be change in: “Water as a sensor and an amplifier: the water mirror approach”.

Line 158: Please define what “activated” bands mean in this type of approach.

Line from 166 to 173: I suggest moving the paragraph above the previous one, around line 155. In way concepts seems more linked.

Line 188: I suggest not to wrap up the paragraph because the presented concepts are strongly related.

Line 207: Please check the consistency of the notation in the parenthesis.

Line from 226 to 229: I suggest removing this sentence because the data processing step seems to be underestimated.

Line 279: Reference number 1 is repeated in few lines (line 279 and 281); please choose only one of them.

Line from 289 to 292: I suggest postponing this paragraph after the two examples, around line 300.

Line 337: The concept presented in this sentence was already stressed enough, I suggest to remove this sentence.

Section 3:

- My main concern is the way in which case studies are presented. In my opinion this part is quite long and not well organised. The applications are mixed with the theoretical concepts creating confusion in the reader; moreover, some examples are presented in few words and for others an entire paragraph is written. So, in my opinion a more schematic explanation of the interesting applications of Aquaphotomics has to be present along the whole section.

- I didn’t understand the title of this paragraph because it seems related to technology while it is focused on applications. I suggest rephrasing it.

Line 525: Figure 9 is referred as Figure 11, please check.

Figure 3 and 4: I suggest presenting a different example of spectra because the figures show consecutive spectra of the same sample (miliQ water) but the concept of “perturbation” is explained in the following section 2.3. In fact, in my opinion this is the right group of spectra for Figure 5.

Figure 7: Please resume the caption reporting only crucial information. 

Table 2: I personally don’t like to see the gap in the Table along the column of “Purpose”; if possible please try to summarise the aim of each work.

Last comment: please pay attention on the presentation of References, they are not equally presented and some crucial info are missing in some papers.

Reviewer 4 Report

Review

The Manuscript ‘Aquaphotomics - From Innovative Knowledge to Integrative Platform in Science and Technology’ aims for comprehensively reviewing the field of aquaphotomics, a discipline of science focused on extracting functional information on the properties of complex samples (e.g. biological samples) from the contributions to their vibrational spectra stemming from water molecules, which are one of the major constituents of these kinds of samples.

It is a complete and comprehensively written review, giving the reader (even not familiar with the details and the most recent achievements done in the field) a good understanding of aquaphotomics. That being said, before this work can be recommended for publication, a number of points need to be clarified/explained further by the Authors, in order to remove some vague or disputable statements as outlined below. These are mostly minor remarks.

1.       The Introduction and Discussion are of high quality, and the information presented is exhaustive, giving a very good understanding of the field. However, perhaps it would be beneficial to revise some of the statements that may be found to be overly vague/unclear. Please consider the following.

“…shaped by all of its components and the surrounding energy”. I would strongly advise against using such vague statements as “surrounding energy”. What kind of “energy” Authors have in mind? Instead, I would advise saying “…shaped by all of its components and the properties of the chemical surrounding as well”, or similar.

Also related, in “energy mirror“, the exact meaning of “energy” needs to be explained.

“Water is the second most abundant molecule in the Universe”, while true, it seems to be irrelevant to aquaphotomics, which as described is focused on investigating biological systems and not interstellar matter.

“This unusual behavior stems from the capacity of water molecules for hydrogen”. I would refrain from stating that the hydrogen-bonding occurring in water is unusual. These two sentences seem to suggest so.

It is recommended to put space between the value and the unit (“1440 nm” instead of “1440nm”). Same, as Authors correctly did for e.g. “5000 ppm”.

It is not true, that temperature perturbation as a source of spectral information is used exclusively in aquaphotomics (l. 300-304).

2.       A major scientific concern arises after reading lines 285-286 “In other words, these proteins prevented bending vibrations in water thus keeping water from creating hydrogen bonds with proteins“. Clarifications are needed, as it is difficult to accept the concept of “preventing vibrations” under any circumstances.

Next, the mechanisms that allegedly connects the two assumptions presented in this sentence  (1. “bending vibrations in water” and 2. “keeping water from creating hydrogen bonds with proteins”) remains unclear.

In a similar manner, how exactly Authors understand “less bending vibrations” (l. 283)?

3.      In Figures 3, 4, and 5 the data plotted and listed in the legend need to be explained.

4.      It would be advisable to clarify if the information presented in Table 1 corresponds to a chemically pure water or not.

5.      An aquagram seems to be an important feature but it is explained only very briefly.
